# Impact of climate change on millet yield under different fertilization levels in three agroecological zones of Niger Republic

Ali Malam Labo Mohamed[1]*, Salack Seyni[2], Siyabusa Mkuhlani[3], Abel Chemura[4], Babacar Faye[5], Suleymane Abdul Kadir[6], Agossou Gadedjisso-Tossou[7], Bindawa. Mansur. Auwalu[8]

1 Institut National de la Recherche Agronomique du Niger, Niamey, Niger, 2 WASCAL, Competence Center Blvd Moammar El-Khadafi, Ouagadougou, Burkina Faso, 3 International Institute of Tropical Agriculture, Nairobi, Kenya, 4 Faculty of Geo-Information Science and Earth Observation, Department of Natural Resources, University of Twente, Enschede, Netherlands, 5 Université du Sine Saloum El Hadj Ibrahima Niass, Kaolack, Senegal, 6 Federal College of Horticulture Dadin Kowa, Dadin Kowa, Gombe State, Nigeria, 7 Climate Hazards Center (CHC), University of California, Santa Barbara, California, United States of America, 8 Department of Agronomy, Bayero University, Kano, Nigeria

* mohamedlabo91@gmail.com

## Abstract

The study investigates the impact of climate change on pearl millet production in Niger, focusing on projected changes in temperature and rainfall. The research uses the CERES-millet model in the DSSAT framework to simulate millet yields under three climate scenarios (SSP1–2.6, SSP3–7.0, and SSP5–8.5) for different time periods: 2015–2044, 2045–2074, and 2075–2100. Five Global Circulation Models (GCMs) with varying climate sensitive were selected for simulations these include (IPSL-CM6A-LR, MPI-ESM1–2-HR, GFDL-ESM4, MRI-ESM2–0, and UKESM1–0-LL). The CERES-Millet model was calibrated using field experiment data collected during the 2021 and 2022 rainy seasons at two locations in Niger Goungoubon (2021 and 2022) and Fandou (2022). Trials were established near irrigation facilities to ensure optimal moisture conditions, with supplemental irrigation applied whenever soil moisture dropped below field capacity. Calibration involved iterative adjustment of key genetic coefficients using observed phenological stages and grain yield to improve model accuracy. The performance of the model was then validated by comparing simulated and observed values of phenology and yield showing good agreement and confirming it reliability. The study found that rising temperatures, particularly under high-emission scenarios, lead to shortened flowering and maturity times, with more pronounced effects towards the end of the century. Additionally, changes in rainfall patterns were expected, with an increase in rainfall projected for the Sahel region. The simulations revealed a consistent decline in millet yields across most scenario, with the greatest yield losses occurring under the SSP5–8.5 scenario in the 2075–2100 period. The results highlight the significant threat that climate change

**Data availability statement:** All data generated or analyzed during this study are available at the Figshare repository, accessible via the following DOI link: https://doi.org/10.6084/m9.figshare.29951645.

**Funding:** The author(s) received no specific funding for this work.

**Competing interests:** No author have competing interests.

poses to pearl millet production, emphasizing the need for adaptive crop management strategies.

## Introduction

Pearl millet (*Pennisetum glaucum*) is a principal cereal crop cultivated extensively across the Sahelian zone, where it serves as a vital component of food security. In this context, Niger stands as the second-largest producer of millet in Africa, following Nigeria, and holds the leading position within the Sahel region. The crop is cultivated on approximately 6.7 million hectares of arable land in Niger, with an estimated mean annual production of 3.7 million metric tons over the period 2019–2023. This output constitutes approximately 62% of the nation's total cereal production, underscoring its agronomic and socio-economic significance.

[1]. Despite, its substantial socio-economic and national importance in Niger, pearl millet yields remain relatively low, averaging less than 1 ton per hectare. This low productivity is primarily attributed to a combination of abiotic stresses, including erratic and insufficient rainfall, elevated temperatures, and inherently poor soil fertility. Additionally, biotic factors such as pests and diseases particularly downy mildew and widespread infestations of parasitic weeds like *Striga* further constrain yield potential.

In addition to these existing constraints, projected climate change in arid and semi-arid tropical regions is expected to further exacerbate the challenges facing millet production. The anticipated increases in temperature, variability in precipitation patterns, and frequency of extreme weather events will likely have adverse effects on crop performance, thereby threatening food security in Niger and across the broader Sahel region [2–4]. As a result, domestic production shortfalls may necessitate increased millet imports from neighboring countries such as Nigeria and Mali to bridge the national supply gap.

Pearl millet is inherently well-adapted to drought conditions, making it suitable for cultivation in arid and semi-arid regions [5]. However, its production is significantly limited by environmental stresses, particularly high temperatures and the variability of rainfall both in terms of low total precipitation and erratic distribution [6–8]. Optimal growth and development of pearl millet occur under temperature regimes of approximately 33 °C during the day and 28 °C at night [9]. A substantial rise in temperature beyond the optimal thresholds of 33 °C during the day and 28 °C at night shortens the crop's growing period, leading to a significant decline in grain yield [10–12]. Climate change projections indicate that rising temperatures in arid and semi-arid regions will further suppress millet productivity [3,4]. This is supported by findings from [7], which project a potential 10% reduction in millet yields across Africa by the year 2050 as a consequence of climate change.

Climate change is a global challenge with far-reaching impacts on agriculture, water resources, energy systems, food security, public health, and natural ecosystems [13]. As a result, it has become a strategic priority for many nations, including those in Niger, the broader Sahel region, and across the African continent [14–16].

In particular, studies focusing on the Sahel and savannah zones of Africa have documented a marked rise in extreme temperatures alongside a decline in precipitation over the past five decades [17,18]. Global average temperatures have risen by approximately 1 °C in recent decades, resulting in notable disruptions to the hydrological cycle [19–22]. Niger, in parallel, is increasingly experiencing extreme weather events characterized by rising temperatures and shifts in rainfall patterns, including a general decline in overall precipitation levels.

The combined effects of these climate-related factors pose a serious threat to global agricultural systems, potentially leading to food insecurity in its various dimensions namely food availability, stability, accessibility, and utilization [23,24]. This underscores the importance of assessing the impacts of climate change on crop growth and development, with particular attention to millet, a staple crop of critical importance in Niger.

The traditional approach for investigating crop responses to agronomic practices such as fertilizer application, sowing dates, and irrigation is through field experiments. However, these studies are often resource-intensive, costly, complex, and difficult to sustain in certain regions or over extended periods, particularly when addressing climate change-related issues. In contrast, crop models serve as valuable tools for evaluating the effects of climate and agronomic management strategies on agricultural outcomes, such as crop yield.

Integrating process-based crop models with general circulation models (GCMs), which forecast future climate data, offers a cost-effective and efficient method for facilitating the adaptation of agronomic crop management to climate change [25,26]. This method has been utilized to evaluate the impacts of both present and future climate changes on crop yields [27,28] and to analyze different crop management strategies [29–31]. The Decision Support Systems for Agrotechnology Transfer Model (DSSAT) [32,33] has been thoroughly tested and applied in multiple regions, demonstrating its effectiveness in quantifying crop impacts and assessing adaptation strategies [27–31].

The model has been extensively used in yield gap assessment, decision support, planning, strategic and operational management, as well as in climate change research [34,35]. Furthermore, [36] examined how maize responds to nitrogen (N) and phosphorus (P) under climate change conditions. The findings indicated that maize yields are projected to decrease by 25%−52% by mid-century and 32%−52% by the end of the century, with consistent patterns across all fertilizer application levels.

However, a more significant decline would occur under RCP8.5, with yield reductions ranging from 32% to 59% in the mid-century scenario and 52% to 69% by the end of the century. [7] assessed the effects of climate change on millet and sorghum production in the Sudanian and Sahelian savannas of West Africa. They developed various climate scenarios, combining precipitation changes from −20% to +20% and temperature increases ranging from +0°C to +6°C. Out of the 35 scenarios examined, 31 showed negative impacts on yields, with reductions of up to 41% in the case of a + 6°C temperature increase and a 20% change in rainfall.

Gaining insight into the historical variability of weather and climate factors influencing agricultural production systems and management practices is crucial for developing effective mitigation and adaptation strategies, as well as preparing for extreme weather events in Niger. There is an even more pressing need to examine the interaction between climate change and critical factors affecting millet yield, such as nitrogen fertilization. Research assessing the impact of climate change on crop production in Niger, particularly through the integration of crop models and future climate projections, remains limited. Previous studies have mainly relied on Phase 3 and Phase 5 of the Coupled Model Intercomparison Project (CMIP), which had inherent limitations that diminished the accuracy and reliability of future projections [37,38].

There is a critical gap in research that incorporates CMIP6, the most recent climate projections available [39]. CMIP6 projections offer higher spatial and temporal resolutions, along with enhanced representations of atmospheric dynamics, oceanic processes, sea ice, land surface interactions, and biogeochemical cycles [40]. Conducting research based on these advanced projections is essential to improve the accuracy and reliability of future climate impact assessments, particularly in the context of agricultural systems in Niger.

 

The primary goal of this study is to deepen the understanding of climate change impacts on rainfed millet yields under different nitrogen fertilization regimes, employing the DSSAT-CERES-Millet model in the Republic of Niger. Specifically, the research aims to assess and quantify the effectiveness of various adaptation strategies to climate change, with an emphasis on enhancing resilience to climate variability and its associated agricultural impacts.

## Materials and methods

### Calibration and validation

To enhance the reliability of simulations generated by the DSSAT-CERES-Millet model, it is essential to perform model calibration and validation using observed experimental data. Calibration typically involves three years of data, while a separate year is reserved for validation. The DSSAT model requires comprehensive datasets, including leaf area index, soil properties, weather conditions, crop management practices, final yields, and associated yield components spanning at least four growing seasons. The soil dataset should encompass both physical and chemical characteristics across multiple depths within the rooting zone profile [32].

Field experiments for model calibration were conducted during the 2021 and 2022 rainy seasons at two locations: Goungoubon (2021 and 2022) and Fandou (2022). Trials were established in proximity to irrigation infrastructure to ensure optimal soil moisture levels, with supplemental irrigation applied whenever soil moisture dropped below field capacity. To maximize the yield potential of the two millet varieties, organic manure was applied at a rate of 10 tons per hectare prior to planting and incorporated into the soil during land preparation. Sowing dates at the Goungoubon site were 14 July 2021 and 16 July 2022. At the Fandou site, sowing was carried out on 14 July 2022, with a target plant population of 30,000 plants per hectare. At planting, nitrogen (N) was applied at a rate of 30 kg N ha$^{-1}$ using urea. Potassium (K) was supplied as Muriate of Potash at a rate of 30 kg $K_2O$ ha$^{-1}$, while phosphorus (P) was applied as Single Super Phosphate at 30 kg $P_2O_5$ ha$^{-1}$. The remaining nitrogen, also at 30 kg N ha$^{-1}$, was applied in the form of urea in two split doses: immediately after thinning and again at the booting stage.

The experiment was laid out in a Randomized Complete Block Design (RCBD) with three replications. Land preparation involved clearing and harrowing prior to sowing. Millet was planted in plots measuring 5 × 4 meters (20 m²), each containing four rows spaced 100 cm apart. Seeds previously treated with the pesticide Calthio (20 g per 5 kg of seed) were sown at a rate of 5–6 seeds per planting hole, with a spacing of 1 meter between holes. Thinning was carried out two weeks after germination, reducing the stand to three plants per hole.

For the validation phase, five field trials were conducted under natural rain-fed conditions. A split-plot experimental design with three replications was employed. The trials took place at Magaria, Tarna-Maradi, and Ndounga in 2020, and at Tarna-Maradi and Ndounga in 2021. The treatments included four sowing dates (15 June, 29 June, 13 July, and 27 July) and two pearl millet varieties: PPBTERRA and SOSAT-C88. PPBTERRA is classified as a medium-maturing variety, while SOSAT-C88 is considered an extra-medium maturing type.

Planting dates were allocated to the main plots, while the two millet varieties were assigned to the subplots. Each subplot consisted of four rows, each 5 meters in length (4 rows × 5 m). Seeds were sown in planting holes spaced 1 meter apart, with 5–6 seeds per hole. Thinning was performed two weeks after emergence, reducing the stand to three plants per hole, thereby achieving a target plant density of 30,000 plants per hectare.

At the time of planting, phosphorus was applied at a rate of 30 kg $P_2O_5$ per hectare using single super phosphate (SSP), while potassium was supplied at 30 kg $K_2O$ per hectare in the form of muriate of potash. Nitrogen was initially applied at 30 kg per hectare as urea shortly after thinning. An additional 30 kg per hectare of nitrogen, also in the form of urea, was applied at the booting stage. Weed and pest management practices were carried out as needed throughout the growing season.

Data were collected on flowering time, maturity, as well as grain yields. Based on these measured parameters, the DSSAT crop model was calibrated and validated. Model validation was conducted using the d-index and root mean square error (RMSE) to assess performance.

This study did not involve human participants, animals, personal data, or research conducted in protected areas, and therefore did not require ethics committee approval under applicable regulations. The research was conducted under the official mandate of the *Institut National de la Recherche Agronomique du Niger*, the national agency responsible for agricultural research in Niger. INRAN reviewed and authorized the study, which was carried out in collaboration with other regional research institutions

## Soil data

Prior to planting, three soil profile pits were excavated at each site. Soil samples were collected from each layer within the profiles and transported to the laboratory for analysis of both biophysical and chemical properties. These analyses included measurements of soil pH (in water), texture, moisture content, bulk density, exchangeable potassium (K), organic matter content, available phosphorus (using the Bray II method), total nitrogen, and cation exchange capacity (CEC). To prepare the data for use in the DSSAT model, SBuild an internal DSSAT soil data formatting tool was employed to convert the soil information into a model-compatible format.

## Input weather data

Weather data used for calibrating and validating the model were sourced from Trans-African Hydro-Meteorological Observatory (TAHMO). The key parameters included daily measurements of rainfall, minimum and maximum temperatures, and solar radiation, covering the period from January 2021 to December 2022. To make the data compatible with the DSSAT model, Weatherman an internal DSSAT weather data conversion tool was used to format the information into a model-ready and readable format.

## Climate change scenarios and yield gap analyses

The Global Circulation Models (GCMs) used in this study were downscaled and bias-corrected within the framework of the Inter-Sectoral Impact Model Intercomparison Project (ISIMIP3b, https://www.isimip.org/). According to Salack et al. (2022), the models were selected based on two main criteria: (i) structural independence in their oceanic and atmospheric components, and (ii) the quality of their process representation, which was rated as fair (for IPSL-CM6A-LR and MPI-ESM1–2-HR) and robust (for GFDL-ESM4, MRI-ESM2–0, and UKESM1–0-LL) according to an informal expert survey.

These GCMs represent a broad range of the CMIP6 ensemble, encompassing three models with low climate sensitivity (GFDL-ESM4, MPI-ESM1–2-HR, MRI-ESM2–0) and two with high climate sensitivity (IPSL-CM6A-LR, UKESM1–0-LL). The ISIMIP3b dataset includes historical simulations from 1850 to 2014, and future projections from 2015 to 2100, provided at a daily time step and 0.5° x 0.5° spatial resolution (Table 1)

For this study, three Shared Socioeconomic Pathways (SSPs) were used to capture a range of potential future conditions:

- **SSP1–2.6**: A low-emissions scenario aligned with the Paris Agreement's goal of limiting global warming to 2°C above pre-industrial levels. It involves declining greenhouse gas (GHG) emissions reaching net-zero by 2050, followed by varying degrees of net negative $CO_2$ emissions.

**Table 1. Description of the global climate models for predicting future climate change under the CMIP6 based projections.**

| Models | Acronym | Institute | Resolution | Reference |
|---|---|---|---|---|
| GFDL-ESM4 | GFDL | Geophysical Fluid Dynamics Laboratory | 1.25°x1.00°; 49 levels | [41] |
| IPSL-CM6A-LR | IPSL | Instityte Pierre-Simon Laplace | N96 (2.5°x1.259°); 79 levels | [42] |
| MPI-ESM1–2-HR | MPI | Max Planck Institute Earth System Model | T127 (0.94°x0.94°); 95 levels | [43] |
| MRI-ESM2–0 | MRI | Moteorological Research Institute | TL159 (~120 km); 80 levels | [44] |
| UKESM1–0-LL | UKE | Met Office Hadley Centre | N96 (1.875°x1.25°); 85 levels | [45] |

- **SSP3–7.0**: A medium-to-high emissions "middle road" scenario with significant regional rivalry and limited climate mitigation.
- **SSP5–8.5**: A high-emissions scenario characterized by intensive fossil fuel use and steep increases in GHG emissions throughout the 21st century [2,40].

The impact assessment was conducted by comparing the simulated yields of two millet varieties under baseline conditions (1983–2012) with projected yields for three future periods: near-term (2015–2044), mid-term (2045–2074), and long-term (2075–2100). These comparisons were made under three different Shared Socioeconomic Pathways (SSPs) and across seven nitrogen (N) fertilization levels: 0, 10, 20, 30, 40, 50, and 60 kg N ha$^{-1}$. To quantify the differences in productivity under various management scenarios, the yield gap was calculated using the following formula.:

$$\Delta \text{Yield} \ (\%) \ = \left( \frac{YieldScenario - YieldBaseline}{YieldBaseline} \right)$$

Where: ΔYield is the change in yield due to climate change, YieldScenario and Yieldbaseline are the yields obtained under scenario and baseline weather conditions, respectively.

## Results

### Calibration and validation results

Table 2 presents key genetic parameters for the growth and development of PPBTERRA and SOSAT-C88. PPBTERRA has a longer juvenile phase, requiring 197 °C days compared to 160 °C days for SOSAT-C88, indicating slower early development. Likewise, the grain-filling period is longer in PPBTERRA (160 °C days) than in SOSAT-C88 (140 °C days), suggesting more time for biomass accumulation or grain filling.

Both cultivars share a critical photoperiod of 12 hours and have equal sensitivity to photoperiod, with a developmental delay of 100 °C days for each hour beyond this threshold. The phyllochron interval, representing thermal time between successive leaf appearances, is the same (43 °C days), as are growth trait coefficients (1.20), indicating similar leaf emergence and tillering rates. Morphologically, PPBTERRA has a slightly larger leaf size scalar (0.40) than SOSAT-C88 (0.35), which may enhance photosynthesis. However, SOSAT-C88 allocates more assimilates to the panicle (partitioning scalar of 1.47) compared to PPBTERRA (0.85), reflecting a greater investment in grain development. These genetic differences suggest that PPBTERRA favors slower development with larger leaves, while SOSAT-C88 prioritizes faster development and allocates more resources to grain production, which could influence yield and adaptation to different environments.

**Table 2. Genetic Parameters for Two Cultivars: PPBTERRA and SOSAT-C88.**

| Parameter | Description | Unit | PPBTERRA | SOSAT-C88 |
|---|---|---|---|---|
| P1 | Thermal time from seedling emergence to end of juvenile phase | °C day | 197.0 | 160.0 |
| P2O | Critical photoperiod above which development slows | hours | 12.00 | 12.00 |
| P2R | Delay in development per hour increase above P2O | °C day | 100.0 | 100.0 |
| P5 | Thermal time from start of grain filling to physiological maturity | °C day | 160.0 | 140.0 |
| G1 | Relative leaf size scalar | unitless | 0.40 | 0.35 |
| G4 | Assimilate partitioning scalar (to panicle) | unitless | 0.85 | 1.47 |
| PHINT | Phylochron interval (thermal time between successive leaf tip appearances) | °C day | 43.00 | 43.00 |
| GT | Growth trait coefficient (e.g., tillering or transition rate) | unitless | 1.20 | 1.20 |
| G5 | Scalar for grain number or panicle size | unitless | 11.0 | 11.0 |

The calibration and validation results of the CERES-millet model demonstrated a strong alignment between observed and simulated days to flowering. For the SOSAT-C88 and PPBTERRA millet cultivars, the model achieved d-index values of 0.78 and 0.80, and RMSE values of 0.80 and 0.50, respectively, indicating good model performance.

The simulated and observed days to maturity showed strong alignment for both cultivars, with d-index values of 0.78 for SOSAT-C88 and 0.80 for PPBTERRA. The root mean square error (RMSE) for days to maturity was notably low only 0.8 days for SOSAT and 0.5 days for PPBTERRA indicating high model accuracy. Similarly, the simulated grain yield closely matched the observed values, with RMSE values of 54.49 kg ha⁻¹ for SOSAT-C88 and 33.3 kg ha⁻¹ for PPBTERRA. This strong agreement was further confirmed by high agreement index values, ranging from 0.95 to 0.96 for SOSAT and PPB-TERRA, respectively Table 3.

The model demonstrated satisfactory performance during validation, with simulated days to flowering closely matching the observed values. The d-index for days to flowering was high, recorded at 0.70 for SOSAT-C88 and 0.85 for PPB-TERRA (Fig 1a), while the RMSE values remained low 3.06 days for SOSAT-C88 and 1.69 days for PPBTERRA. Similarly, the model accurately predicted days to maturity, showing strong agreement with observed data. For SOSAT-C88, the RMSE was 2.27 days with a d-index of 0.87, and for PPBTERRA, the RMSE was 1.83 days with a d-index of 0.86 (Fig 1b). Additionally, simulated grain yields aligned well with observations for both varieties, with a d-index of 0.98 and RMSE of 96 for SOSAT, and a d-index of 0.83 with RMSE of 130.08 for PPBTERRA (Fig 1c)

**Table 3. Calibration results of the DSSAT CERES–Millet model for days to flowering, maturity, and grain yields for two cultivars in Niger.**

| Cultivar | Days to flowering | | | | Days to maturity | | | | Grain Yield | | | |
|---|---|---|---|---|---|---|---|---|---|---|---|---|
| | Sim | Obs | Statistics | | Sim | Obs | Statistics | | Sim | Obs | Statistics | |
| | Days | | RMSE | d–index | Days | | RMSE | d–index | kg ha-1 | | RMSE | d–index |
| SOSAT-C88 | 60 | 60 | 0.82 | 0.78 | 80 | 79 | 0.81 | 0.78 | 2034 | 2014 | 54.49 | 0.95 |
| | 61 | 62 | | | 82 | 82 | | | 1970 | 1990 | | |
| | 61 | 61 | | | 82 | 82 | | | 2013 | 2007 | | |
| PPBTERRA | 52 | 52 | 0.5 | 0.8 | 72 | 72 | 0.5 | 0.88 | 2481 | 2394 | 33.3 | 0.96 |
| | 53 | 54 | | | 73 | 74 | | | 2173 | 2138 | | |
| | 53 | 54 | | | 73 | 74 | | | 2216 | 2207 | | |

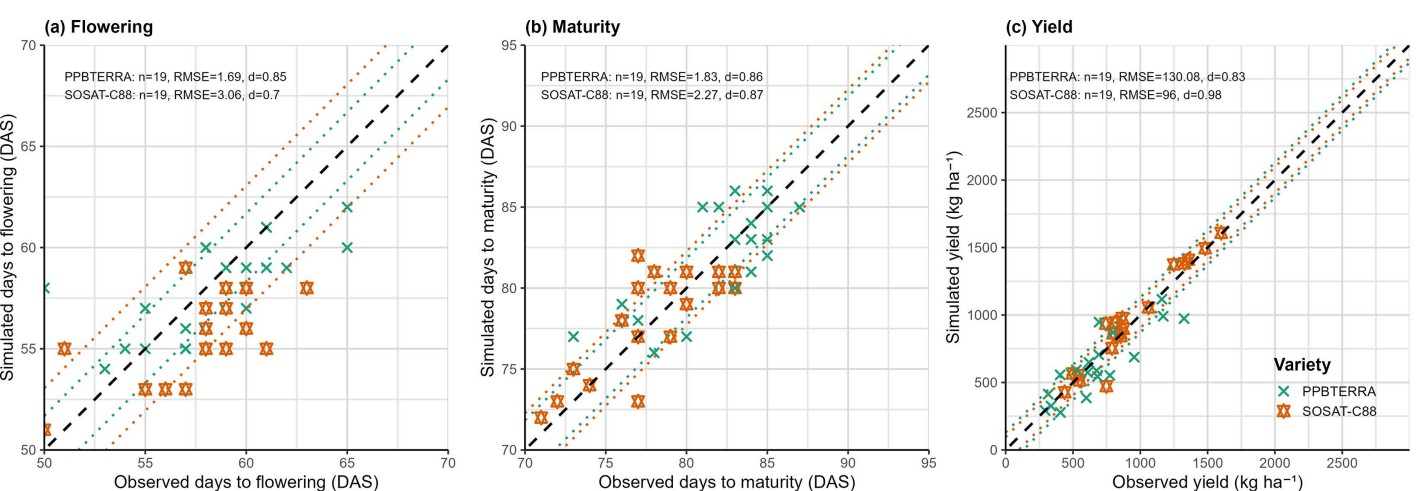

**Fig 1. Comparison of the days simulated and observed to flowering (a), physiological maturity (b) and millet grain yields (c) for Niger.**

## Projected changes in temperature and seasonal rainfall

At Dabala, maximum temperature is projected to increase progressively across all climate change scenarios (SSP1–2.6, SSP3–7.0, and SSP5–8.5) and future time periods (2015–2044, 2045–2074, and 2075–2100). During the near-term period (2015–2044), projected warming remains relatively moderate, with temperature increases ranging from 0.23 °C, according to the MPIE model under SSP3–7.0, to 2.22 °C, as projected by the UKSM model under SSP1–2.6. This early-stage warming reflects the onset of climate change impacts at the local scale (Fig 2)

In the mid-century period (2045–2074), the temperature rise becomes more pronounced. The most significant increase is projected under the SSP5–8.5 scenario, where the UKSM model indicates a warming of 4.10 °C, suggesting an intensification of heat-related stress during this period. By the end of the century (2075–2100), substantial warming is projected across all scenarios, with the most severe increases occurring under the high-emission SSP5–8.5 pathway. Temperature increases range from 0.72 °C to 6.72 °C, with the UKSM model again projecting the highest rise (Fig 2).

During the 2015–2044 period, the maximum temperature at Gotheye is projected to increase across all scenarios. The smallest rise of 0.21 °C is projected by the MPIE model under SSP1–2.6, while the highest, 1.50 °C, is expected under SSP5–8.5 according to the UKSM model. In the 2045–2074 period, warming becomes more pronounced, with projections ranging from 1.02 °C (MPIE, SSP1–2.6) to 4.16 °C (UKSM, SSP5–8.5). By the end of the century (2075–2100), a substantial increase is projected, reaching 6.60 °C under the high-emission SSP5–8.5 scenario as estimated by the UKSM model (Fig 2).

At Yakoye Tounga, during the 2015–2044 period, projected increases in maximum temperature range from 0.17 °C under SSP3–7.0 with the MPIE model to 1.35 °C under the same scenario with the UKSM model. In the 2045–2074 period, the warming intensifies, varying between 0.14 °C (MPIE, SSP5–8.5) and 3.65 °C (UKSM, SSP5–8.5). By the end of the century (2075–2100), projections show further temperature escalation, with increases spanning from 0.70 °C under SSP1–2.6 (MPIE) to 6.12 °C under SSP5–8.5 (UKSM) (Fig 2).

At Dabala, during the near-future period (2015–2044), the projected increase in minimum temperature ranges from 0.65°C with the MPIE model under SSP3–7.0 to 2.22°C with the UKSM model under SSP1–2.6. In the mid-future period (2045–2074), the increase spans from 1.12°C with MPIE under SSP1–2.6 to 3.48°C with the IPSL model under

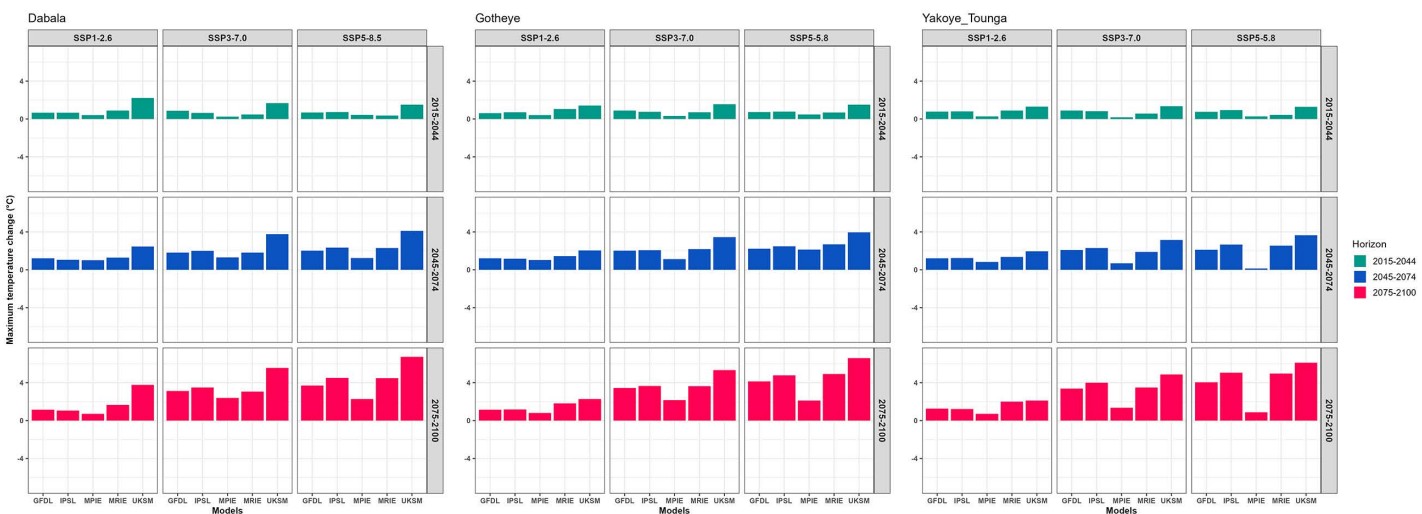

**Fig 2. Change in maximum air temperature (° C) from the baseline period (1983–2012) from future climate change represented by five GCMs under SSP1-2.6, SSP3-7.0 and SSP5-8.5 by 2015-2044, 2045-2074 and 2075-2100 in Sudan, Sudan Sahel and Sahel Agroecological Zones (AEZ) in the Niger Republic.**

SSP5–8.5. By the end of the century (2075–2100), the projected rise ranges from 0.82°C with MPIE under SSP1–2.6 to 6.21°C with IPSL under SSP5–8.5 (Fig 3).

At Gotheye, near-future projections show an increase from 0.67°C with MPIE under SSP1–2.6 to 1.77°C with UKSM under SSP5–8.5. In the mid-century, the rise is projected between 1.20°C (MPIE, SSP1–2.6) and 4.16°C (UKSM, SSP5–8.5) (Fig 3).

At Yakoye Tounga, during the near future, the minimum temperature is projected to increase from 0.51°C (MPIE, SSP3–7.0) to 1.35°C (UKSM, SSP3–7.0). In the mid-century under SSP5–8.5, the increase ranges from 1.01°C (GFDL) to 3.49°C (UKSM) (Fig 3).

Overall, the temperature increase is more pronounced during the mid- and end-of-century periods, especially under the SSP3–7.0 and SSP5–8.5 scenarios.

For rainfall, projections from all models indicate an overall increase at both Dabala and Gotheye under the SSP1–2.6, SSP3–7.0, and SSP5–8.5 scenarios. At Dabala, the most substantial increase is expected under SSP5–8.5, followed by SSP3–7.0, particularly during the late-century period (2075–2100). Gotheye exhibits a similar trend, with rainfall amounts increasing progressively across the scenarios (Fig 4).

In contrast, rainfall projections for Yakoye Tounga are more inconsistent, showing fluctuating patterns across all future timeframes. Although most models suggest an increase in rainfall, the GFDL model forecasts a decline under all three scenarios SSP1–2.6, SSP3–7.0, and SSP5–8.5 for both mid-century and end-century periods. A decrease in rainfall is also projected specifically under SSP1–2.6 during 2045–2074 and 2075–2100 (Fig 4).

### Impact on the phenology of millet

For the PPBTERRA cultivar, a climate change-driven decline in days to flowering is projected across Dabala, Gotheye, and Yakoye Tounga under all scenarios and future periods. The largest reductions occur at Yakoye Tounga, followed by Gotheye and Dabala. Specifically, the MRIE model under the SSP5–8.5 scenario for 2075–2100 predicts the greatest shortening of flowering time by 8.97 days, while the smallest decrease of 0.26 days is projected at Dabala during 2015–2044 under SSP3–7.0 by the MPIE model (Fig 5).

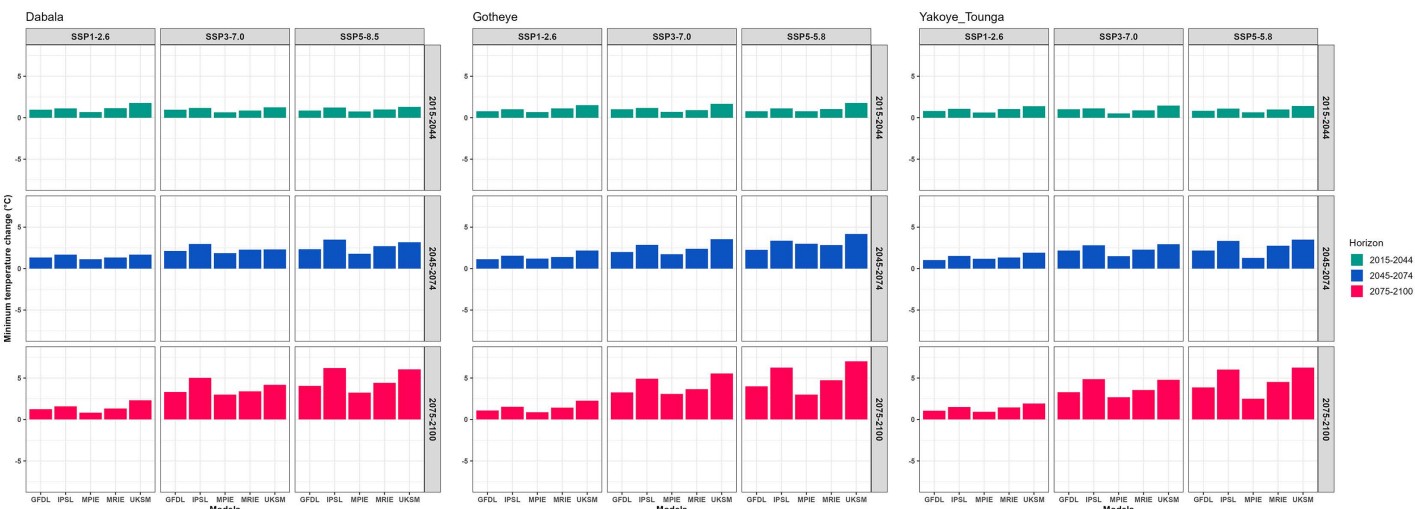

**Fig 3. Change in minimum air temperature (° C) from the baseline period (1983–2012) from future climate change represented by five GCMs under SSP1-2.6, SSP3-7.0 and SSP5-8.5 by 2015-2044, 2045-2074 and 2075-2100 in Sudan, Sudan Sahel and Sahel Agroecological Zones (AEZ) in the Niger Republic.**

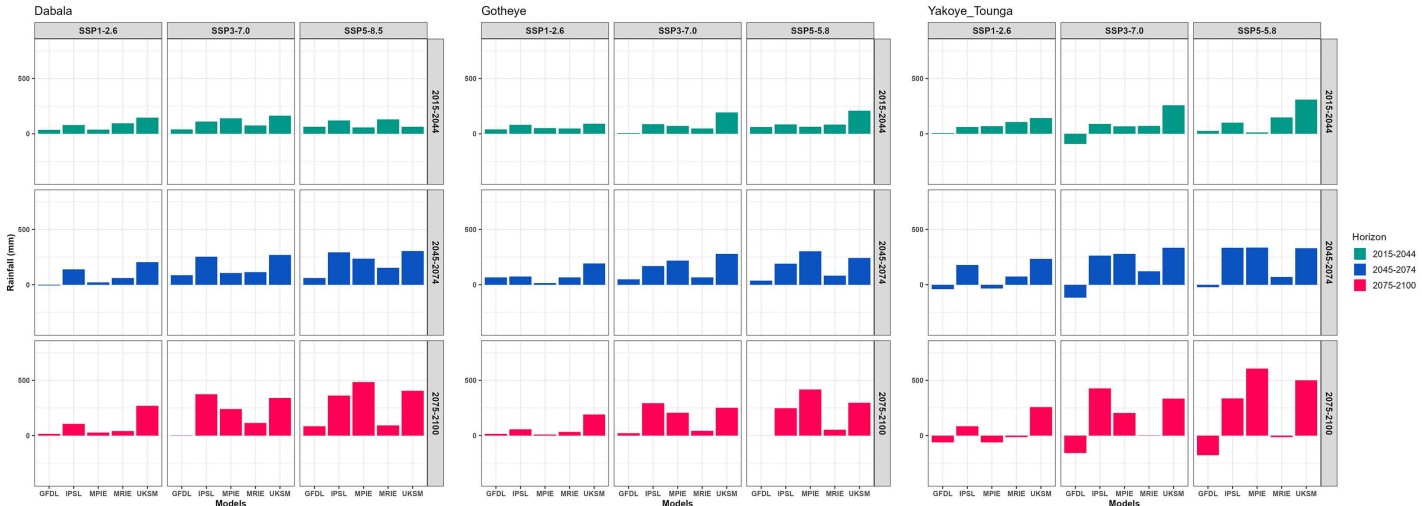

**Fig 4. Change in rainfall (mm) from the baseline period (1983–2012) from future climate change represented by five GCMs under SSP1-2.6, SSP3-7.0, and SSP5-8.5 by 2015-2044, 2045-2074, and 2075-2100 in Sudan, Sudan Sahel and Sahel Agro-ecological Zones (AEZs) in the Niger Republic.**

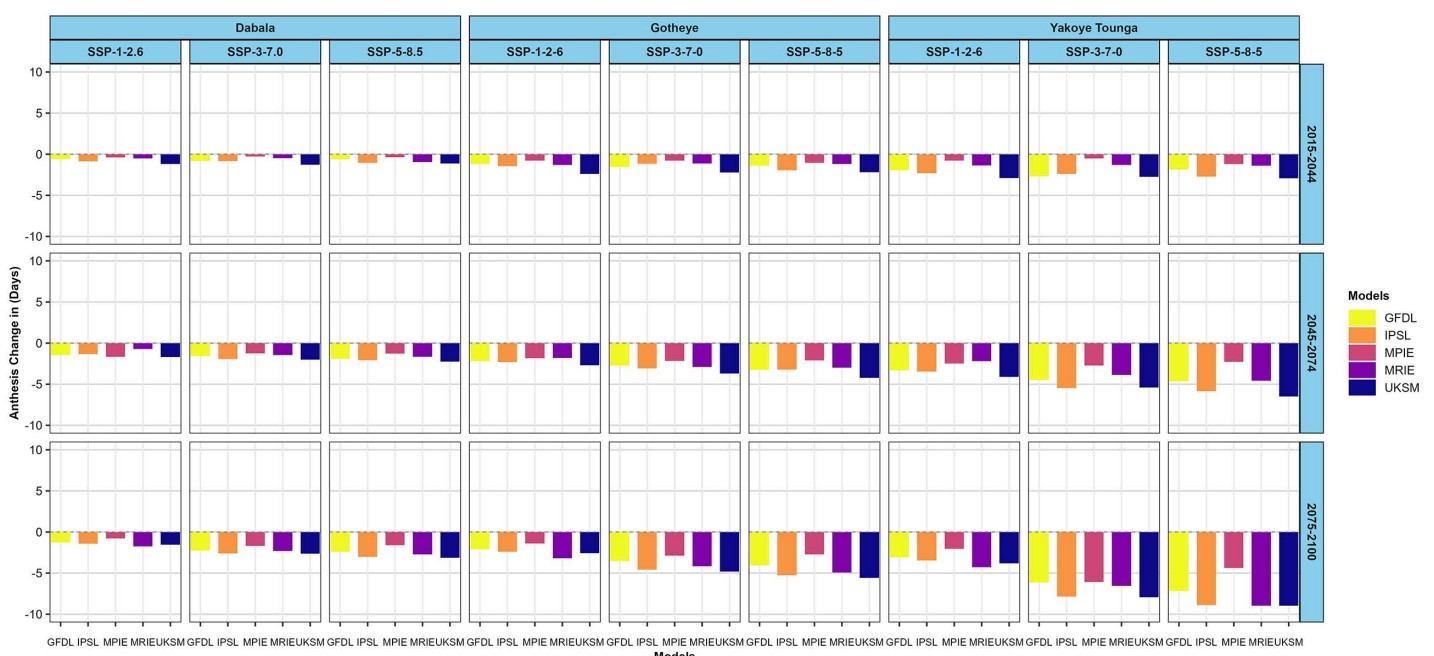

**Fig 5. Changes in millet (PPBTERRA) anthesis in days across different nitrogen levels in the near future (2015−2044), mid-avenir (2045−2074) and end-future (2075−2100) for three scenarios (SSP1-2.6, SSP3-7.0 and SSP5-5.8) compared to the baseline scenario in Sudan, Sudan Sahel and Sahel Agro-ecological zones (AEZ) in the Niger Republic at 50 N kg ha$^{-1}$.**

Similarly, the SOSAT-C88 cultivar is expected to exhibit earlier flowering across all climate scenarios and periods at Dabala. The most significant advancement 6.96 days earlier flowering is forecast by the UKSM model under SSP5–8.5 for 2075–2100 (Fig 6) At Gotheye, days to flowering are also expected to shorten, with reductions ranging from 0.43 days

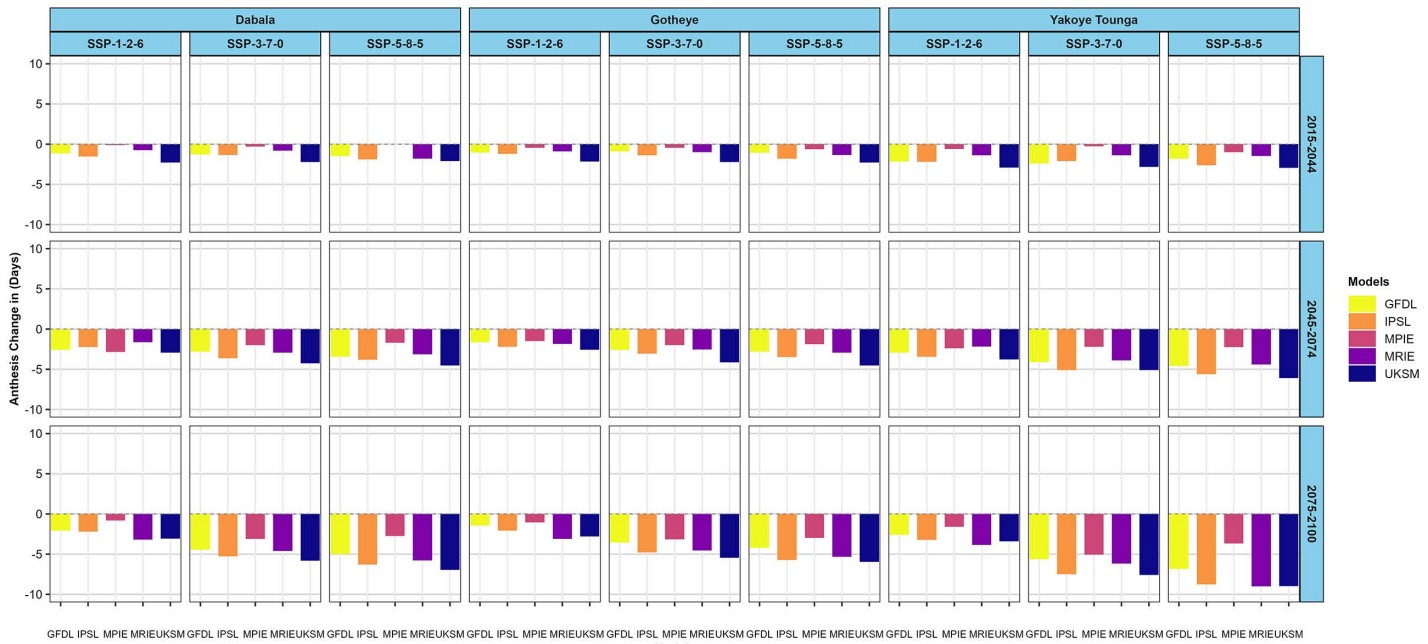

**Fig 6. Changes in millet (SOSAT-C88) anthesis in days across different nitrogen levels in the near future (2015−2044), mid-avenir (2045−2074) and end-future (2075−2100) for three scenarios (SSP1-2.6, SSP3-7.0 and SSP5-5.8) compared to the baseline scenario in Sudan, Sudan Sahel and Sahel Agro-ecological zones (AEZ) in the Niger Republic at 50 N kg ha⁻¹.**

(MPIE, SSP3–7.0, 2015–2044) to 5.96 days (UKSM, SSP5–8.5, 2075–2100), reflecting variability across models and scenarios.

Regarding maturity, for the PPBTERRA cultivar, the largest climate change-induced reduction at Dabala is 10.31 days (IPSL, SSP5–8.5, 2075–2100), while the smallest decline of 0.3 days occurs in 2015–2044 under SSP3–7.0. At Gotheye, maturity duration is projected to shorten by up to 8.60 days (UKSM, SSP5–8.5, 2075–2100), with minimal decrease of 0.83 days (MPIE, SSP1–2.6, 2015–2044). Yakoye Tounga shows the greatest reduction of 13.75 days (UKSM, SSP5–8.5, 2075–2100), but uniquely exhibits an increase of 2.37 days in days to maturity under SSP1–2.6 during 2015–2044 MPIE (Fig 7).

For the SOSAT-C88 cultivar, climate change is projected to shorten maturity by as much as 10.53 days at Dabala (UKSM, SSP5–8.5, 2075–2100), with minimal reduction of 0.3 days under SSP3–7.0 (MPIE, 2015–2044). At Gotheye, the largest and smallest decreases are 9.37 days (UKSM, SSP5–8.5, 2075–2100) and 0.5 days (MPIE, SSP3–7.0, 2015–2044), respectively. Yakoye Tounga experiences the most substantial shortening of 13.32 days (MRIE, SSP5–8.5, 2075–2100), with the smallest decline of 0.86 days under SSP3–7.0 during 2015–2044 (Fig 8)

### Impact of climate change on millet yields

Millet yield projections for the PPBTERRA cultivar demonstrate notable spatial and temporal variability across the agro-ecological zones of Dabala, Gotheye, and Yakoye Tounga. These variations are influenced by socio-economic pathways (SSP1–2.6, SSP3–7.0, SSP5–8.5), time horizons (2015–2044, 2045–2074, 2075–2100), climate models (GFDL, IPSL, MPIE, MRIE, UKSM), and nitrogen fertilizer application rates (0–60 N kg ha⁻¹).

In Dabala, projected yield changes during the near-future period (2015–2044) are generally small across most climate scenarios and models. Notable exceptions include the GFDL model, which forecasts yield increases under SSP3–7.0 and

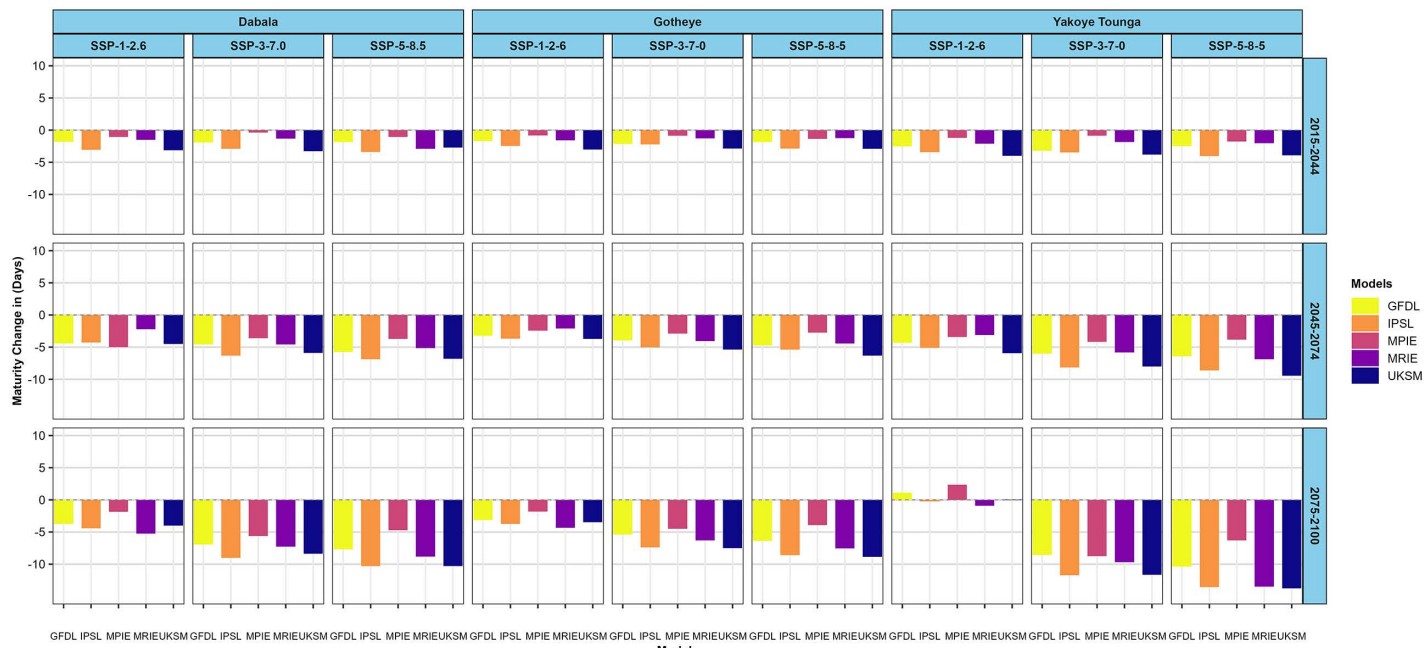

**Fig 7. Changes in millet (PPBTERRA) maturity in days across different nitrogen levels in the near future (2015−2044), mid-avenir (2045−2074) and end-future (2075−2100) for three scenarios (SSP1-2.6, SSP3-7.0 and SSP5-5.8) compared to the baseline scenario in Sudan, Sudan Sahel and Sahel Agro-ecological zones (AEZ) in the Niger Republic at 50 N kg ha⁻¹.**

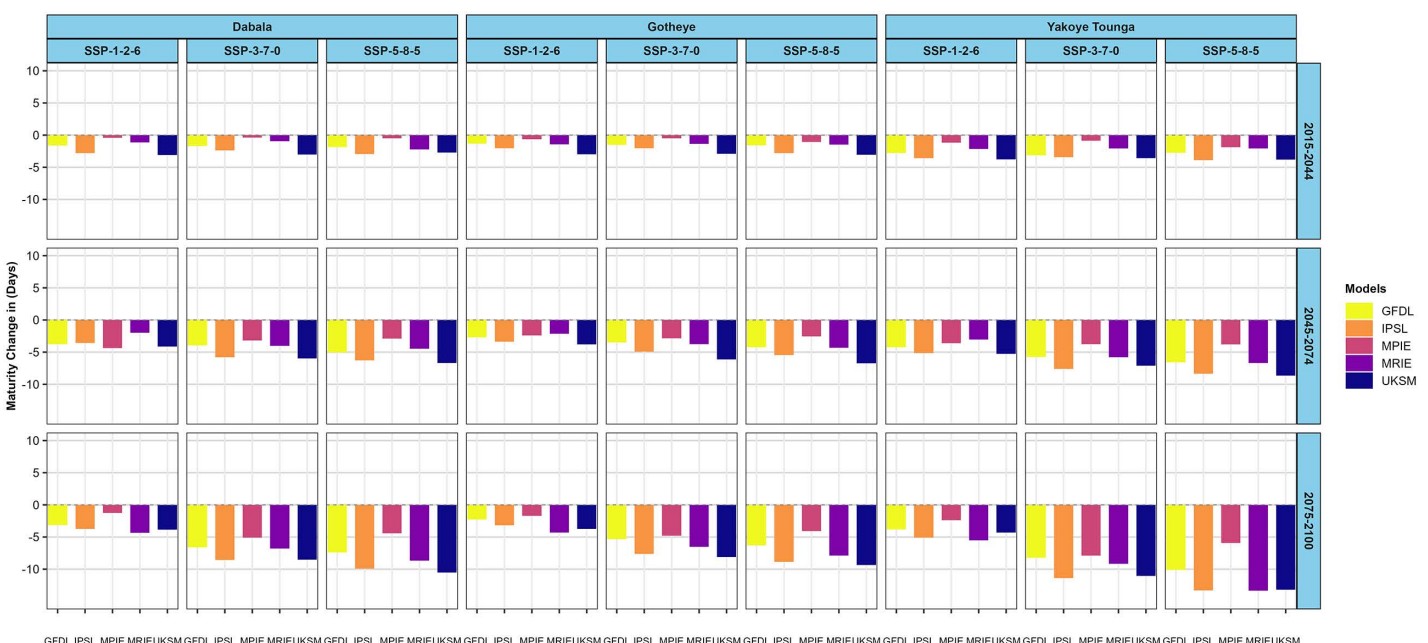

**Fig 8. Changes in millet (SOSAT-C88) maturity in days across different nitrogen levels in the near future (2015−2044), mid-avenir (2045−2074) and end-future (2075−2100) for three scenarios (SSP1-2.6, SSP3-7.0 and SSP5-5.8) compared to the baseline scenario in Sudan, Sudan Sahel and Sahel Agro-ecological zones (AEZ) in the Niger Republic at 50 N kg ha⁻¹.**

SSP5–8.5 at higher nitrogen levels, particularly at 60 N kg ha⁻¹. Other models and scenarios indicate minimal fluctuations, suggesting low sensitivity of millet yields to climatic and fertilizer changes during this time (Fig 9).

By mid-century (2045–2074), yield declines become more evident, particularly under SSP3–7.0 and SSP5–8.5, with losses intensifying at higher nitrogen application levels. While most models indicate decreasing yields, the MRIE model occasionally projects slight increases under SSP1–2.6. Toward the end of the century (2075–2100), significant yield reductions dominate across nearly all models and scenarios. An exception is the MPIE model, which projects a yield increase at 60 N kg ha⁻¹ under SSP5–8.5. Despite this isolated improvement, the general trend shows that elevated nitrogen inputs are associated with deeper yield losses, and even under the low-emission SSP1–2.6 scenario, reductions persist, though at a lower intensity. The most pronounced losses are projected by the UKSM, MRIE, and IPSL models (Fig 9)

In Gotheye, projected changes in millet yield during the near-term period remain minor. Slight increases are observed under SSP1–2.6 and SSP3–7.0 with the MPIE model, and under SSP5–8.5 with the GFDL model at lower nitrogen levels. Overall, yield responses remain neutral or slightly positive during this early period. By mid-century, the MPIE model continues to project yield improvements under SSP3–7.0 and SSP5–8.5 across all nitrogen levels, with the highest gains occurring at 60 N kg ha⁻¹. In contrast, the UKSM, MRIE, and IPSL models forecast greater yield declines under the same scenarios. Under SSP1–2.6, the GFDL model projects the most significant losses. By the end of the century, yield reductions intensify across most models and scenarios, particularly under SSP5–8.5 at higher nitrogen application rates (Fig 9).

In Yakoye Tounga, yield projections are consistently negative across all scenarios, time periods, and nitrogen levels. In the near-future horizon, reductions are already evident and tend to increase with higher nitrogen inputs. This declining trend intensifies by mid-century, particularly under SSP3–7.0 and SSP5–8.5, with the greatest losses observed at nitrogen rates of 40–60 N kg ha⁻¹. Even under SSP1–2.6, yields continue to decline, although, to a lesser extent. By the end of the

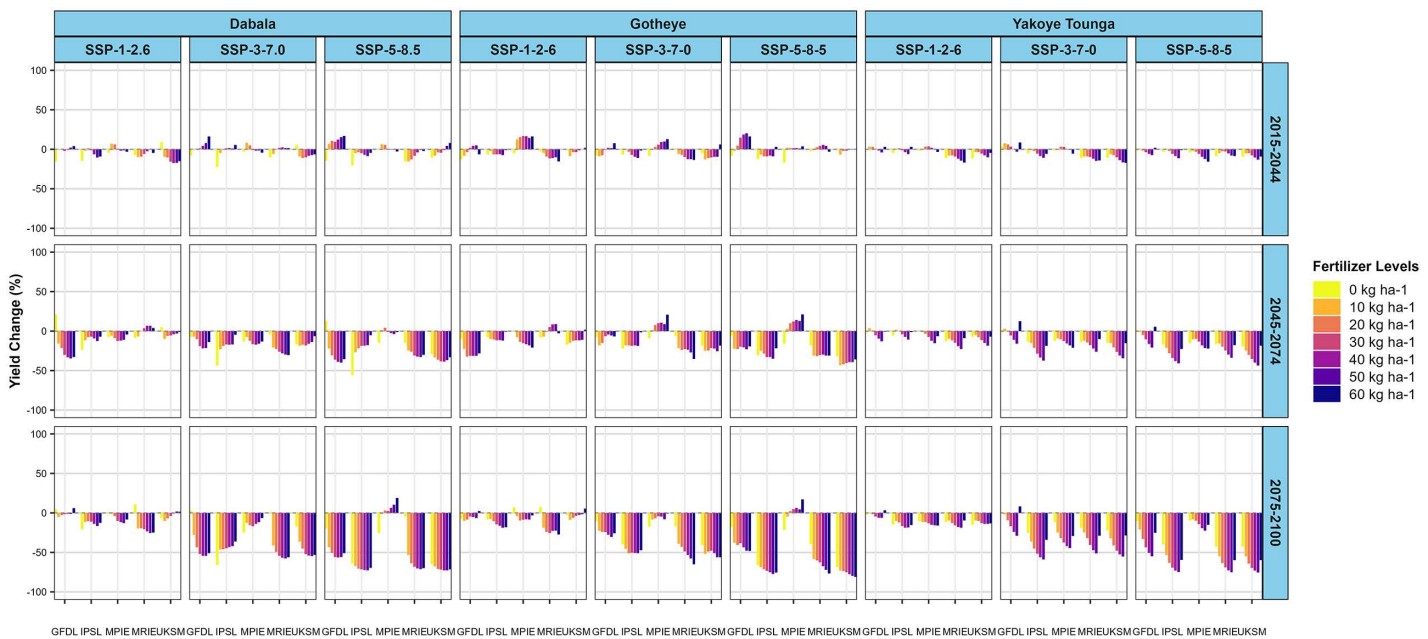

**Fig 9. Changes in millet (PPBTERRA) yields (%) across different nitrogen levels in the near future (2015−2044), mid-avenir (2045−2074) and end-future (2075−2100) for three scenarios (SSP1-2.6, SSP3-7.0 and SSP5-5.8) compared to the baseline scenario in Sudan, Sudan Sahel and Sahel Agro-ecological zones (AEZ) in the Niger Republic.**

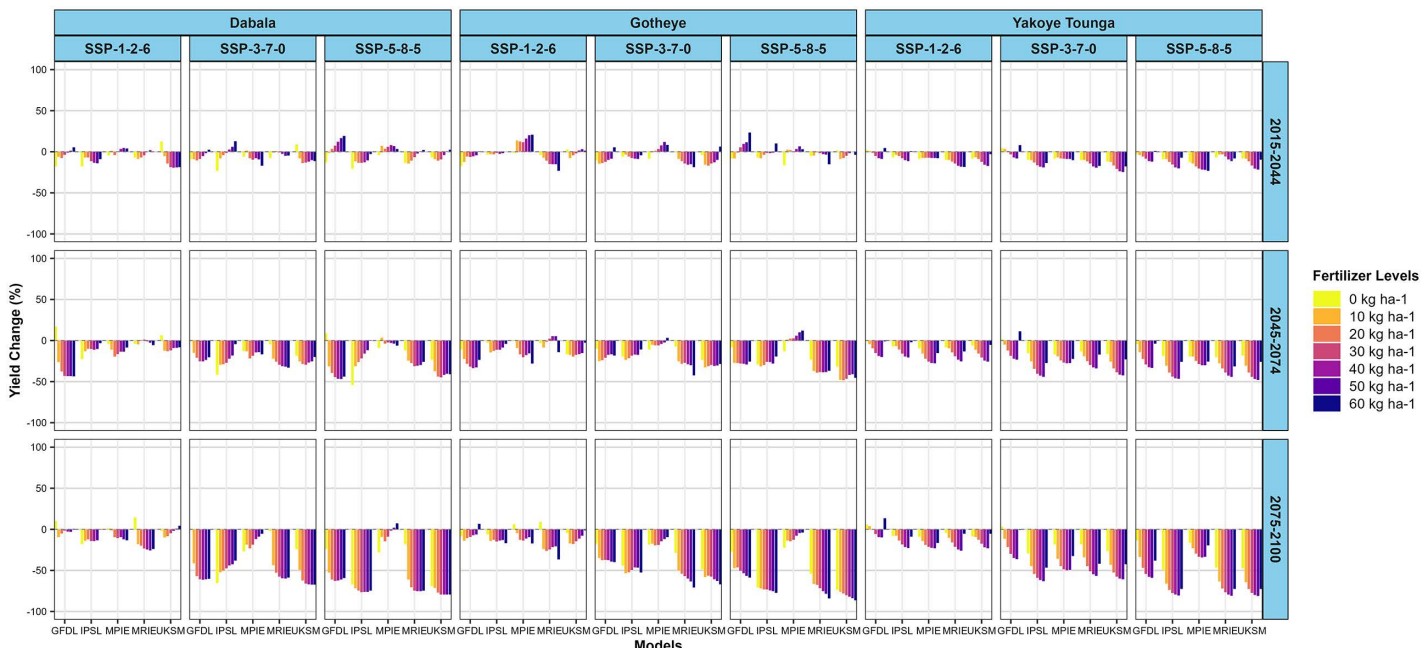

**Fig 10. Changes in millet (SOSAT-C88) yields (%) across different nitrogen levels in the near future (2015−2044), mid-avenir (2045−2074) and end-future (2075−2100) for three scenarios (SSP1-2.6, SSP3-7.0 and SSP5-5.8) compared to the baseline scenario in Sudan, Sudan Sahel and Sahel Agro-ecological zones (AEZ) in the Niger Republic.**

century, all models forecast severe yield deterioration, especially at high nitrogen levels, with no scenario indicating any yield improvement. This consistent downward trend across models, scenarios, and nitrogen levels highlights the heightened vulnerability of this zone to future climate change impacts (Fig 9).

For the SOSAT-C88 cultivar, millet yields at both Dabala and Gotheye are generally projected to decline across all climate scenarios (SSP1–2.6, SSP3–7.0, and SSP5–8.5) and future time periods. The most pronounced yield reductions are consistently associated with higher nitrogen application levels. However, a few exceptions are observed, particularly during the near-future period (2015–2044). At Dabala, a slight increase in yield is projected under SSP5–8.5 with the GFDL model. At Gotheye, yield improvements are noted under SSP1–2.6 and SSP3–7.0 with the MPIE model, and under SSP5–8.5 with the GFDL model. Additionally, during the mid-century period (2045–2074), the MPIE model continues to project a slight yield increase at Gotheye under SSP5–8.5 (Fig 10).

## Discussion

In Niger, agriculture is one of the sectors most vulnerable to climate change, contributing approximately 40.6% to the country's overall economic output. The effects of climate change have resulted in severe losses, primarily due to increased variability in climate patterns, especially rainfall, both over time and across regions. This variability has contributed to recurring cycles of drought and flooding. Utilizing crop models, such as DSSAT, offers a cost-effective and relatively straightforward method for enhancing our understanding of climate vulnerability, assessing impacts, and exploring potential adaptation strategies.

The DSSAT CERES-millet model showed strong performance in simulating phenological stages (anthesis and maturity) and grain yield during both calibration and validation. Phenology calibration is a critical step in crop modeling, as it allows the model to accurately reflect how growth stages and yield respond to variations in temperature and rainfall. These growth phases are particularly sensitive to such climate factors [46]. Additionally, all statistical indices indicated a good

match between observed and simulated grain yields. These outcomes are consistent with previous findings [47], confirming that the CERES-millet model is suitable for evaluating climate change impacts on millet production. The model relies on a wide range of parameters for effective calibration and validation [32], and its accuracy could be further improved by integrating additional data such as leaf area index and soil moisture.

For the minimum and maximum temperatures, all the GCMs showed an increase in temperature across all the scenarios and time periods. These results are consistent with the report of [48] showing an increase in rainfall with differences increased under RCP 8.5 compared to RCP 4.5 and towards the inner Sahel than in the Sudano Sahelian and Sudanian agroclimatic zones. These results are consistent with the [48] showing an increase in precipitation with differences heightened under RCP 8.5 compared to RCP 4.5, and toward the inner Sahel than in the Sudano-Sahelian and Sudanian agroclimatic zones.

Rainfall is projected to rise across all locations, scenarios, and time periods, except at Yakoye Tounga, where rainfall is expected to follow a variable pattern. These results align with those of [49], who observed precipitation increases of up to 30% under the RCP 8.5 scenario in Birni N′ Konni (Sudano-Sahelian zone).

Crop phenology, particularly stages like flowering and maturation, is highly influenced by climate variables such as temperature and atmospheric $CO_2$ concentrations. The timing of these stages is critical for determining overall crop productivity. Flowering (anthesis) and maturity are especially significant, as they govern the grain-filling period, a key determinant of final yield. The study found a noticeable reduction in the number of days to flowering and maturity for both cultivars, with the most significant decrease observed under elevated $CO_2$ levels, specifically in the high-emission scenario SSP5–8.5 towards the end of the century (2075–2100), compared to other scenarios.

Rising temperatures tend to accelerate plant development, leading to shorter growing seasons, reduced grain-filling periods, and lower biomass accumulation. Likewise, elevated $CO_2$ levels can cause crops to flower and mature earlier, further limiting the duration needed for full grain development. This interaction between temperature and $CO_2$ creates complex effects that may contribute to reduced yields and biomass under future climate change scenarios. Although increased $CO_2$ concentrations under various RCP scenarios may influence crop phenology, experimental results show mixed effects, with no clear or consistent trends. In general, higher temperatures during the growing season tend to accelerate phenological development, resulting in shortened vegetative and reproductive phases.

Yield projections under climate change reveal a consistent downward trend across most Global Climate Models (GCMs), regardless of the time frame, nitrogen application rate, crop variety, or scenario. The most significant yield reductions are anticipated under the high-emission scenario SSP5–8.5 by the end of the century (2075–2100). Interestingly, despite many models forecasting increased rainfall, crop yields still decline, indicating that higher precipitation does not necessarily lead to improved agricultural output.

This yield reduction is likely linked to rising maximum and minimum temperatures. Higher temperatures may offset the potential benefits of increased rainfall by enhancing evaporative water loss and increasing crop water demand. In addition to climatic stress, non-climatic factors also contribute to reduced productivity. Numerous studies have established that elevated temperatures lead to greater evapotranspiration, which negatively impacts crop yields [7,50,51].

For cereal crops, in particular, the growth rate tends to increase linearly with temperature from a base temperature (Tbase) up to an optimal threshold. However, beyond this optimal temperature, growth rates decline sharply and may cease altogether at temperatures around 40°C [52,53]. These findings highlight the complex relationship between climate variables and agricultural performance, emphasizing the vulnerability of crop yields to rising temperatures.

In line with this, Sultan et al. [7] found that a projected 4°C temperature rise could result in a 28% reduction in yields for short-cycle crop varieties, with losses potentially reaching 40% for long-cycle varieties. Moreover, yield declines are more severe under the high-emission scenario SSP5–8.5 compared to SSP3–7.0. Supporting this trend, research by [54] and [55] identified average temperature as a key factor negatively impacting maize yields an observation that aligns with the results of the present study.

An increase in temperature can accelerate the rate of crop development, shorten the growth period, and ultimately lead to lower yields [56–58]. Under the RCP8.5 scenario, maize yields are expected to decline more significantly than under RCP4.5, primarily due to higher temperatures and reduced time for crop growth. According to [59], future climate change may cause crops to reach maturity more rapidly, resulting in a potential 30% reduction in grain yield. This is mainly due to the shortening of crop life cycles and faster growth rates, which increase respiration losses, limit biomass accumulation, and reduce final yields.

Furthermore, nitrogen application simulations showed a consistent decline in projected yields across all climate scenarios and timeframes for both crop varieties. The negative impact of rising temperatures on yield was more severe when nitrogen was applied at higher rates across all study locations. This suggests that increased use of mineral fertilizers is not a suitable adaptation strategy to address the effects of climate change. These findings are consistent with the conclusions of [36].

In many low-input systems across the Sahel, soils are inherently nutrient-poor, and farmers often lack access to sufficient mineral fertilizers. Under such conditions, the response to nitrogen can be constrained by other soil limitations such as organic matter and moisture availability [60–62]. Consequently, yield gains from nitrogen application alone may not be realized, especially under climate stress. This highlights the importance of considering nitrogen-stressed and low-fertility scenarios in future impact assessments and adaptation planning, which is more reflective of the realities in semi-arid smallholder systems.

One promising adaptation strategy is the integrated application of organic manure and mineral fertilizers, which has been shown to enhance soil structure, water retention, and nutrient cycling [63,64]. Such integrated soil fertility management (ISFM) practices improve nutrient use efficiency and increase resilience against climatic stressors. Evidence from the West African Sahel suggests that combining organic inputs, such as compost or animal manure, with reduced rates of mineral fertilizers can maintain or even increase yields under low rainfall and high-temperature conditions [65,66].

Additionally, earlier flowering and maturity, particularly under high-emission scenarios like SSP5–8.5, shorten the vegetative and reproductive phases, thereby narrowing the window for nutrient uptake. This has critical implications for fertilizer application strategies. Traditional blanket fertilizer schedules may become less effective under future climate conditions. Evidence from similar agroecological zones suggests that revised nutrient management strategies such as micro-dosing, split applications, and synchronization with phenological stages can enhance efficiency under shortened crop cycles [67,68]. Tailoring fertilizer application to these phenological shifts could improve nutrient use efficiency and partially offset some of the climate-induced yield losses.

## 5. Conclusions

The DSSAT model has demonstrated strong effectiveness in accurately simulating both millet phenology and yield across diverse agro-ecological zones in Niger and the broader Sahel region. This enhances confidence in assessing the impacts of climate change on millet production, which was the central objective of this study. The results align with global trends indicating rising temperatures, with the most significant increases projected under high-emission scenarios and towards the end of the century. Likewise, future changes in rainfall patterns are anticipated. Interestingly, while global projections often suggest a decline in rainfall, this study forecasts an increase in seasonal rainfall for the study area highlighting that the effects of climate change on precipitation are highly location-specific. This underlines the importance of conducting detailed spatial and temporal analyses of rainfall projections.

Despite the projected increase in rainfall, it is not sufficient to counteract the adverse effects of rising temperatures, which continue to drive reductions in crop yield. Moreover, the yield reductions were more pronounced under high nitrogen application rates, suggesting that reliance on mineral fertilizers may not be a viable adaptation strategy under future warming conditions. These findings emphasize the need for the alternative and more integrated adaptations approaches.

In response, this recommends a combination of strategies, including the adjustment of sowing windows to match climatic conditions, the development or selection of climate-resilient ideotypes with improved stress tolerance, and the realignment of fertilizer application timing to better match changing crop phenology. Where feasible, supplemental irrigation during critical growth stages may also help stabilize yields in the face of increasing rainfall variability. These actionable insights provide a foundation for developing climate-smart millet production systems that enhance resilience in the Sahel.

## Supporting information

**S1 File. Inclusivity-in-global-research-questionnaire.**
(DOCX)

## Author contributions

**Conceptualization:** Ali Malam Labo Mohamed, Salack Seyni, Bindawa. Mansur. Auwalu.

**Data curation:** Ali Malam Labo Mohamed, Salack Seyni, Abel Chemura, Babacar Faye.

**Formal analysis:** Ali Malam Labo Mohamed, Siyabusa Mkuhlani, Abel Chemura, Babacar Faye, Suleymane Abdul Kadir, Agossou Gadedjisso-Tossou.

**Methodology:** Ali Malam Labo Mohamed, Siyabusa Mkuhlani, Agossou Gadedjisso-Tossou.

**Software:** Ali Malam Labo Mohamed.

**Supervision:** Ali Malam Labo Mohamed, Salack Seyni, Bindawa. Mansur. Auwalu.

**Writing – original draft:** Ali Malam Labo Mohamed.

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
