## [Decision Letter · Decision Letter 0]

24 Jul 2025

PONE-D-25-15778Impact of climate change on millet yield under different fertilization levels in three agroecological zones of Niger RepublicPLOS ONE

Dear Dr. Mohamed,

Thank you for submitting your manuscript to PLOS ONE. After careful consideration, we feel that it has merit but does not fully meet PLOS ONE’s publication criteria as it currently stands. Therefore, we invite you to submit a revised version of the manuscript that addresses the points raised during the review process.

We look forward to receiving your revised manuscript.

Kind regards,

Ahmed Kheir, PhD

Academic Editor

PLOS ONE

**Journal Requirements:**

1. When submitting your revision, we need you to address these additional requirements. Please ensure that your manuscript meets PLOS ONE's style requirements, including those for file naming. The PLOS ONE style templates can be found at https://journals.plos.org/plosone/s/file?id=wjVg/PLOSOne_formatting_sample_main_body.pdf and https://journals.plos.org/plosone/s/file?id=ba62/PLOSOne_formatting_sample_title_authors_affiliations.pdf 2. Please include a complete copy of PLOS’ questionnaire on inclusivity in global research in your revised manuscript. Our policy for research in this area aims to improve transparency in the reporting of research performed outside of researchers’ own country or community. The policy applies to researchers who have travelled to a different country to conduct research, research with Indigenous populations or their lands, and research on cultural artefacts. The questionnaire can also be requested at the journal’s discretion for any other submissions, even if these conditions are not met.  Please find more information on the policy and a link to download a blank copy of the questionnaire here: https://journals.plos.org/plosone/s/best-practices-in-research-reporting. Please upload a completed version of your questionnaire as Supporting Information when you resubmit your manuscript. 3. In your Methods section, please provide additional information regarding the permits you obtained for the work. Please ensure you have included the full name of the authority that approved the field site access and, if no permits were required, a brief statement explaining why. 4. Please note that PLOS ONE has specific guidelines on code sharing for submissions in which author-generated code underpins the findings in the manuscript. In these cases, we expect all author-generated code to be made available without restrictions upon publication of the work. Please review our guidelines at https://journals.plos.org/plosone/s/materials-and-software-sharing#loc-sharing-code and ensure that your code is shared in a way that follows best practice and facilitates reproducibility and reuse. 5. Please include your tables as part of your main manuscript and remove the individual files. Please note that supplementary tables (should remain/ be uploaded) as separate "supporting information" files. 6. Please provide a complete Data Availability Statement in the submission form, ensuring you include all necessary access information or a reason for why you are unable to make your data freely accessible. If your research concerns only data provided within your submission, please write "All data are in the manuscript and/or supporting information files" as your Data Availability Statement. 7. Please amend either the title on the online submission form (via Edit Submission) or the title in the manuscript so that they are identical. 8. We note that Figure 1 in your submission contain map images which may be copyrighted. All PLOS content is published under the Creative Commons Attribution License (CC BY 4.0), which means that the manuscript, images, and Supporting Information files will be freely available online, and any third party is permitted to access, download, copy, distribute, and use these materials in any way, even commercially, with proper attribution. For these reasons, we cannot publish previously copyrighted maps or satellite images created using proprietary data, such as Google software (Google Maps, Street View, and Earth). For more information, see our copyright guidelines: http://journals.plos.org/plosone/s/licenses-and-copyright. We require you to either present written permission from the copyright holder to publish these figures specifically under the CC BY 4.0 license, or remove the figures from your submission: a. You may seek permission from the original copyright holder of Figure 1 to publish the content specifically under the CC BY 4.0 license.   We recommend that you contact the original copyright holder with the Content Permission Form (http://journals.plos.org/plosone/s/file?id=7c09/content-permission-form.pdf) and the following text:“I request permission for the open-access journal PLOS ONE to publish XXX under the Creative Commons Attribution License (CCAL) CC BY 4.0 (http://creativecommons.org/licenses/by/4.0/). Please be aware that this license allows unrestricted use and distribution, even commercially, by third parties. Please reply and provide explicit written permission to publish XXX under a CC BY license and complete the attached form.” Please upload the completed Content Permission Form or other proof of granted permissions as an "Other" file with your submission. In the figure caption of the copyrighted figure, please include the following text: “Reprinted from [ref] under a CC BY license, with permission from [name of publisher], original copyright [original copyright year].” b. If you are unable to obtain permission from the original copyright holder to publish these figures under the CC BY 4.0 license or if the copyright holder’s requirements are incompatible with the CC BY 4.0 license, please either i) remove the figure or ii) supply a replacement figure that complies with the CC BY 4.0 license. Please check copyright information on all replacement figures and update the figure caption with source information. If applicable, please specify in the figure caption text when a figure is similar but not identical to the original image and is therefore for illustrative purposes only.The following resources for replacing copyrighted map figures may be helpful: USGS National Map Viewer (public domain): http://viewer.nationalmap.gov/viewer/The Gateway to Astronaut Photography of Earth (public domain): http://eol.jsc.nasa.gov/sseop/clickmap/Maps at the CIA (public domain): https://www.cia.gov/library/publications/the-world-factbook/index.html and https://www.cia.gov/library/publications/cia-maps-publications/index.htmlNASA Earth Observatory (public domain): http://earthobservatory.nasa.gov/Landsat:
http://landsat.visibleearth.nasa.gov/USGS EROS (Earth Resources Observatory and Science (EROS) Center) (public domain): http://eros.usgs.gov/#Natural Earth (public domain): http://www.naturalearthdata.com/ 9. If the reviewer comments include a recommendation to cite specific previously published works, please review and evaluate these publications to determine whether they are relevant and should be cited. There is no requirement to cite these works unless the editor has indicated otherwise. 

Reviewers' comments:

Reviewer's Responses to Questions

**Comments to the Author**

1. Is the manuscript technically sound, and do the data support the conclusions?

Reviewer #1: Yes

Reviewer #2: Yes

2. Has the statistical analysis been performed appropriately and rigorously? 

Reviewer #1: No

Reviewer #2: Yes

3. Have the authors made all data underlying the findings in their manuscript fully available?

Reviewer #1: Yes

Reviewer #2: Yes

4. Is the manuscript presented in an intelligible fashion and written in standard English?

Reviewer #1: Yes

Reviewer #2: Yes

5. Review Comments to the Author

**Reviewer #1: ** The study accolades a relevant and timely model-based assessment of pearl millet production in Niger’s Sahel region. Given that pearl millet is a major staple crop and most of the country’s production comes from this region, assessing its response under current and future climate scenarios is both scientifically and practically important. The crop already suffers from low yield, partly due to climatic stress and biotic pressures. Therefore, simulating Genotype × Environment × Management interactions offers a promising approach to identify management strategies that could help close the yield gap. The manuscript is generally well written—especially the abstract and introduction, which provide a good narrative flow. However, the Materials and Methods, as well as Results and Discussion sections, could be improved substantially.

Abstract:

The abstract is reasonable. However, please briefly mention the calibration procedure (location and time), and specify how many and which GCMs were used.

General:

It was difficult to note down detailed comments as the manuscript lacks line numbers—please include them in future versions.

Introduction:

It would be more informative to start by stating where the Sahel region is located and why it is favourable for millet cultivation (e.g., spring temperatures, rainfall distribution). Also, please clarify whether the 62% figure refers to total millet production or total cereal production in the region.

Materials and Methods:

Please include a table summarising model parameters, especially if multiple cultivars were used. Provide relevant cultivar information. Replace the term “evaluation” with “validation” throughout the manuscript. Also, instead of just writing “DSSAT,” use the full model name—DSSAT-CERES-Millet—consistently.

Results and Discussion:

Discuss the limitations of the study more explicitly. It would be useful to include how future improvements can be made. The impact of temperature rise and phenological shifts is well documented in existing literature; the novelty here could be strengthened by assessing nitrogen stress under low-fertility scenarios, which are common in these environments. This would be more meaningful than reiterating known phenology shifts. Also, early flowering and maturity will require a shift in fertiliser application strategies—this should be discussed clearly. You may consider drawing on previous studies that have addressed nitrogen limitations in similar agroecological zones.

Conclusion:

The conclusion should be expanded with more actionable insights—e.g., shifting sowing windows, identifying or breeding suitable ideotypes, and aligning fertiliser management with changing crop phenology.

Figures and Tables:

Please add standard deviation or error bars to figures. If possible, conduct ANOVA to test for significant differences among GCM outputs. For Figure 9, since biomass data was not used for model calibration, I suggest omitting biomass-related outputs from the model application results.

**Reviewer #2:**  Firstly, I appreciate the effort has been spent to produce such a valuable study. Contents and tools are up to date and mostly important in those types of studies. I appreciate also that the author's team has produced the work without receiving specific funding, which is by itself a great success.

I have some minor corrections as follows:

1- According to the submitted manuscript standards of the journal, you would need to number the lines all over the manuscript with a continuous numbering. This would make our job as reviewers.

2- In the abstract, in line 6, you need to substitute the word "reduced" to "shortened". In line no. 9, you need to put the word "scenarios" instead of "models". In line no. 11, you need to correct the phrase to "Climate Change's threat".

3- in the conclusions, in line no. 1, you need to correct the phrase to "has been proven". In line no. 4, you need to correct a phrase to "on the study".

With all my best wishes of success and continuous progress.

6. PLOS authors have the option to publish the peer review history of their article (what does this mean? ). If published, this will include your full peer review and any attached files.

**Do you want your identity to be public for this peer review?** For information about this choice, including consent withdrawal, please see our Privacy Policy .

Reviewer #1: No

Reviewer #2: No

---

## [Editor Report · Decision Letter 1]

15 Sep 2025

PONE-D-25-15778R1Impact of climate change on millet yield under different fertilization levels in three agroecological zones of Niger RepublicPLOS ONE

Dear Dr.  Mohamed,

Thank you for submitting your manuscript to PLOS ONE. After careful consideration, we feel that it has merit but does not fully meet PLOS ONE’s publication criteria as it currently stands. Therefore, we invite you to submit a revised version of the manuscript that addresses the points raised during the review process.

We look forward to receiving your revised manuscript.

Kind regards,

Ahmed Kheir, PhD

Academic Editor

PLOS ONE

Journal Requirements:

Additional Editor Comments:

**Reviwer#1**The study accolades a relevant and timely model-based assessment of pearl millet production in Niger’s Sahel region. Given that pearl millet is a major staple crop and most of the country’s production comes from this region, assessing its response under current and future climate scenarios is both scientifically and practically important. The crop already suffers from low yield, partly due to climatic stress and biotic pressures. Therefore, simulating Genotype × Environment × Management interactions offers a promising approach to identify management strategies that could help close the yield gap. The manuscript is generally well written—especially the abstract and introduction, which provide a good narrative flow. However, the Materials and Methods, as well as Results and Discussion sections, could be improved substantially.

Abstract:

The abstract is reasonable. However, please briefly mention the calibration procedure (location and time), and specify how many and which GCMs were used.

General:

It was difficult to note down detailed comments as the manuscript lacks line numbers—please include them in future versions.

Introduction:

It would be more informative to start by stating where the Sahel region is located and why it is favourable for millet cultivation (e.g., spring temperatures, rainfall distribution). Also, please clarify whether the 62% figure refers to total millet production or total cereal production in the region.

Materials and Methods:

Please include a table summarising model parameters, especially if multiple cultivars were used. Provide relevant cultivar information. Replace the term “evaluation” with “validation” throughout the manuscript. Also, instead of just writing “DSSAT,” use the full model name—DSSAT-CERES-Millet—consistently.

Results and Discussion:

Discuss the limitations of the study more explicitly. It would be useful to include how future improvements can be made. The impact of temperature rise and phenological shifts is well documented in existing literature; the novelty here could be strengthened by assessing nitrogen stress under low-fertility scenarios, which are common in these environments. This would be more meaningful than reiterating known phenology shifts. Also, early flowering and maturity will require a shift in fertiliser application strategies—this should be discussed clearly. You may consider drawing on previous studies that have addressed nitrogen limitations in similar agroecological zones.

Conclusion:

The conclusion should be expanded with more actionable insights—e.g., shifting sowing windows, identifying or breeding suitable ideotypes, and aligning fertiliser management with changing crop phenology.

Figures and Tables:

Please add standard deviation or error bars to figures. If possible, conduct ANOVA to test for significant differences among GCM outputs. For Figure 9, since biomass data was not used for model calibration, I suggest omitting biomass-related outputs from the model application results.

 Reviewer#2

Firstly, I appreciate the effort has been spent to produce such a valuable study. Contents and tools are up to date and mostly important in those types of studies. I appreciate also that the author's team has produced the work without receiving specific funding, which is by itself a great success.

I have some minor corrections as follows:

1- According to the submitted manuscript standards of the journal, you would need to number the lines all over the manuscript with a continuous numbering. This would make our job as reviewers.

2- In the abstract, in line 6, you need to substitute the word "reduced" to "shortened". In line no. 9, you need to put the word "scenarios" instead of "models". In line no. 11, you need to correct the phrase to "Climate Change's threat".

3- in the conclusions, in line no. 1, you need to correct the phrase to "has been proven". In line no. 4, you need to correct a phrase to "on the study".

With all my best wishes of success and continuous progress.

---

## [Author Response · Author response to Decision Letter 2]

18 Sep 2025

All the comments have been addressed

---

## [Editor Report · Decision Letter 2]

22 Sep 2025

Impact of climate change on millet yield under different fertilization levels in three agroecological zones of Niger Republic

PONE-D-25-15778R2

Dear Dr. Ali

We’re pleased to inform you that your manuscript has been judged scientifically suitable for publication and will be formally accepted for publication once it meets all outstanding technical requirements.

Kind regards,

Ahmed Kheir, PhD

Academic Editor

PLOS ONE
---

## [Editor Report · Acceptance letter]

PONE-D-25-15778R2

PLOS ONE

Dear Dr. Mohamed,

I'm pleased to inform you that your manuscript has been deemed suitable for publication in PLOS ONE. Congratulations! Your manuscript is now being handed over to our production team.

Kind regards,

on behalf of

Dr. Ahmed Kheir

Academic Editor

PLOS ONE